# Pediatric Acute Myeloid Leukemia—Past, Present, and Future

**DOI:** 10.3390/jcm11030504

**Published:** 2022-01-19

**Authors:** Dirk Reinhardt, Evangelia Antoniou, Katharina Waack

**Affiliations:** Pediatrics III, Department of Pediatric Hematology, Oncology and Stem Cell Therapy, University Children’s Hospital, University Duisburg, 45147 Essen, Germany; Evangelia.Antoniou@uk-essen.de (E.A.); katharina.waack@uk-essen.de (K.W.)

**Keywords:** acute myeloid leukemia, children, treatment, prognosis

## Abstract

This review reports about the main steps of development in pediatric acute myeloid leukemia (AML) concerning diagnostics, treatment, risk groups, and outcomes. Finally, a short overview of present and future approaches is given.

## 1. Introduction

The treatment of pediatric acute myeloid leukemia (AML) is a success story in improving prognosis. Whereas, in the 1980s, almost all children suffering from AML died, today, up to 75% of the children survive. However, this is only feasible in a well-structured setting of comprehensive diagnostics, intensive therapy, and effective supportive care. This has been achieved by the cooperative study groups in Europe, North America, and Japan. By contrast, even within Europe, the prognosis of children with AML shows an unacceptable level of inequality of survival rates, ranging from less than 50% to 80% [1]. 

The incidence of pediatric AML is about seven per million, with only minor differences between continents or countries. The malignant blasts originate from early hematopoietic progenitors as an evolution from (pre-)leukemia stem cells. External/environmental factors could explain only a tiny percentage. In addition, predisposing syndromes or germline mutations are associated with less than 10% of pediatric AML. During childhood and adolescence, infants less than two years old and adolescents have the highest incidence. Whereas MLL-rearranged leukemia dominates during infancy, the frequency of core-binding leukemia (CBL) and AML associated with mutations, such as NPM 1 or FLT3-ITD, increases by age [2].

Except for acute promyeloblastic leukemia (APL), improved survival has been achieved by using long-known conventional drugs, mainly cytarabine and anthracyclines. Scheduling risk group stratification, modifications of allogeneic stem cell transplant (alloHSCT), and management of complications allowed for curing most children (Figure 1) [3,4]. 

The only new approach within the most recent 20 years was the targeted CD33 antibody gemtuzumab ozogamicin (GO), which showed some advantages in at least one randomized trial, but GO is only approved in North America [5]. 

The successful therapy of APL with all-trans retinoic acid and arsenic-trioxide is a rare example of curing leukemia targeting specifically leukemia-inducing molecular mechanisms and eradicating the leukemic stem cell [5,6]. 

Liposomal drug formulation of daunoribiicn allowed treatment intensification without increasing toxicities, but this has disappeared due to economic reasons, such as a limited pediatric market. The actual approach with a liposomal nanoscale co-formulation of cytarabine and daunorubicin seems to be promising but needs confirmatory trials in pediatric AML and marketing approval thereof. 

Although stem cell transplantation is still an unspecific treatment option with severe acute and long-term side effects, combined with a better risk group stratification, alloHSCT significantly improved survival in children with high-risk (HR) AML [7,8]. 

There are a broad number of new compounds explicitly targeting signaling pathways. However, it is not confirmed in children to what extent these approaches will contribute to curing or be able to reduce toxicities by allowing reduction of treatment intensity of conventional drugs. 

A fast-growing field are immune and cellular therapies, which show promising results in preclinical and early phase clinical trials (mainly in adults).

## 2. Past

Although pediatric AML has been described since the 1900s, a formal classification was established, such as in adults, in 1976 by the French-American-British (FAB)-Classification [9]. There are already six subtypes of AML that have been established and described. The regular introduction of immunophenotyping modified this morphology and cytochemistry-based classification during the 1990s [10,11,12,13]. In the WHO classification in 2001, a shift from morphology to a primarily genetically-based classification has been released and continuously extended. 

Until 1968, the remission rate was inferior, and the median survival was about 1.5 months [14]. With the implementation of an intensified block therapy, including cyclophosphamide and cytarabine, a survival of 9.5 months was achieved, in 1976 [15].

Since 1975 the first clinical trials for pediatric patients with AML were initiated [16,17,18]. Cooperative Study groups have been established: AIEOP (Associazione Italiana di Ematologia e Oncologia Pediatrica), AML-BFM (Berlin, Frankfurt, Münster), NOPHO (Nordic Society for Pediatric Hematology and Oncology), MRC (Medical Research Council), and EORTC (European Organization of Research and Treatment of Cancer), CCG (Childhood Cancer Group), POG (Pediatric Oncology Group; merged in 2000 to COG (Children’s Oncology Group); and SJCRH (St. Jude Children’s Research Hospital). Whereas the MRC conducted combined pediatric and adult trials [19], the AML-BFM 78 study (1978–1983) was a pure pediatric trial to examine the application of two to six courses of daunorubicin, cytarabine, and 6-thioguanine (DAT) after a first induction of the same course [17]. In the AML 10 trial of the MRC group, the comparison of etoposide with thioguanine, as randomization from ADE versus DAT, showed no significant difference [20]. Significant progress in pediatric AML was made by the AML-BFM 83 trial, based on the introduction of block-scheduling. Within this trial, the favorable risk groups of AML with t(8; 21) and, inv(16) have been identified and confirmed in children. The same favorable cytogenetic criteria were confirmed in the MRC AML 10. As adverse characteristics, -5, -7, del(5q), abn(3q), and complex karyotype were documented [20,21].

Although most patients were defined just by morphology, due to the high correlation between FAB M2 with Auer rods and t(8;21), the presence of atypical eosinophils in FAB M4 Eo with inv(16) or, later, FAB M2 and NPM1-mutated AML allowed event-free survival (EFS) rates of 70% and more [22,23]. Interestingly, despite further intensification, this has not changed until today. In the LAME90/91 study, patients were classified into two groups dependent on whole blood count (WBC) and cytogenetics [24]. The standard risk group included t(8;21), inv(16), t(15;17) (defined as FAB M2, M4, and M3), and also patients with <100 000/μL leukocytes, initially. The AML BFM Study Group described hyperleukocytosis as a poor prognostic marker but was not used for risk stratification [25]. 

Within the 20th century, the relevance of anthracycline analogs has been discussed. Randomized trials, such as AML-BFM 98 and MRC 12/15, tested idarubicin versus daunorubicin or mitoxantrone. Significant achievements were the identification of idarubicin as the most effective anthracycline if applied in a 1 to 5 conversion rate compared to daunorubicin [26,27]. 

In the NOPHO-93, all the patients initially underwent the same treatment (ATEDox), but, dependent on the response after the first induction, an extra AM (cytarabine, Mitoxantrone) induction was recommended [28]. Those patients with excess blasts after the AM course received an HA2E course. 

The tested antileukemic drug in the NOPHO 2004 trial was GO and did not show any significant effect on the recurrence rate of leukemia or overall survival (OS) [29]. An extra criterium of this study was the presence of mixed lineage leukemia (MLL) rearrangements other than t(9;11), suppoeritng for the first time the independent involvement of cytogenetics in the risk stratification. During this study, especially in 2009, the criteria for the high-risk (HR) patients were restricted to the poor response [7]. 

The St. Jude AML02 Study stratified the patients into two subgroups according to morphologic and genetic characteristics. Patients were randomized to receive daunorubicin (50 mg/m^2^ on days 2, 4, and 6) and etoposide (100 mg/m^2^ on days 2–6), and high-dose cytarabine (3 g/m^2^ every 12 hours) [30]. 

The Japanese AML99 study (JPLSG) implemented a risk stratification of three groups. The initial stratification was made for the low-risk patients, and including the HR group criteria of WBC (>100.000 µL) and the age of the patients (<2 years). Additionally, the response after the first induction and the karyotype led to the allocation of the patients in the final stratification group. An alloHSCT was indicated only for the intermediate and HR groups, especially for the last group; a “not familiar” donor was suggested [31].

In the AML05 study (JPLSG), the reduced cumulative anthracycline dose (<300 mg/m^2^) was tested in the low and intermediate-risk patients. At the same time, 50% of the etoposide dose was used in the AML99 protocol. A higher incidence of relapse was noticed, but the OS was not influenced [32]. 

In the early 2000s, international cooperative projects, analyzing a larger cohort of patients, finally defined further prognostic factors [4]. This also led to a harmonization of risk group definition worldwide. All major study groups agreed on AML with t(8;21) and inv(16) as a favorable prognostic group that could be cured with chemotherapy only [23,31]. There are still controversies about the definition and post-remission therapy of the patients belonging to the intermediate-risk group. Some groups recommend alloHSCT, while others stick to chemotherapy. In addition, the HR group was not finally defined, hence remaining heterogenous. In addition, in that period, large treatment groups failed to demonstrate the advantage of alloHSCT in the HR group [33,34,35]. 

### 2.1. Hematopoietic Stem Cell Transplantation

AlloHSCT was introduced in the 1980s as post-consolidation therapy in pediatric AML. Although effective in some cases, it never achieved the status as a general standard in contrast to adults. This is explained by the relevant side and long-term effects and the effective chemotherapy in children, which already allowed in the 1990s a long-term survival of about 60% [36,37]. Finally, the improved risk group stratification allowed the identification of those children who benefit. Nevertheless, it is an ongoing process to identify the pediatric AML subgroups who finally benefit from alloHSCT. This includes all associated issues, such as donor selection, prevention and treatment of graft versus host disease, management of virus reactivation, and immune reconstitution.

Between the mid-1980s and the 1990s, progress in pediatric AML was limited. Autologous HSCT or alloHSCT from a matched sibling donor have been introduced to the therapy [38,39]. Although there was a reduction of relapses, this could not be translated to improved OS. Transplant-related mortality counterbalanced the potentially increased antileukemic effect [40]. Considering the significantly higher risk of post-transplant late sequelae, long-lasting controversies about the relevance and importance of alloHSCT in first complete remission occurred [33,40,41,42,43,44]. 

In the AML-BFM 87 study and the MRC AML 12 trial, the response on day 15 after the first induction was added as a criterium for stratification [27,45]. Whereas alloHSCT was not generally recommended in the AML-BFM trial, the British study group limited alloHSCT to the intermediate and poor-risk group [27]. 

### 2.2. CNS Prophylaxis and Treatment

A monotherapy with intrathecal cytarabine was used as prophylactic therapy in the CCG-2891 and as a treatment twice a week in a total of six doses for the central nervous system (CNS) positive patients [46]. In Italy, the VAPA protocol, the first conducted study in children, also included monotherapy with cytarabine with 12 doses for all included patients [47]. Triple intrathecal therapy with methotrexate, cytarabine, and hydrocortisone was administered in the MCR AML10 and NOPHO 2004 twice a week by CNS positive patients until cerebrospinal fluid (CSF) clearance. The difference was the clearance duration after the extra intrathecal inductions, one and two weeks, respectively, for the MRC and NOPHO treatment groups [29]. 

In the AML-BFM-87 study, prophylactic irradiation was randomized. However, an increased rate of relapses in children without cranial irradiation led to a premature stop [45]. Since then, the AML-BFM studies have included cranial irradiation as a mandatory treatment for all patients, except those who received alloHSCT [48]. In contrast, the CCG and MRC group applied cranial irradiation only if the CSF was not cleared after the intrathecal therapy [35,49]. The standard therapy in the AML02 study of the JPLSG included triple intrathecal therapy in each course but no prophylactic cranial irradiation [31]. Based on the more intensive and CNS-effective chemotherapy, since 2012 the AML-BFM group no longer applied prophylactic cranial irradiation. Only in patients with CNS involvement is irradiation still recommended [50].

To summarize, different intrathecal therapies, such as monotherapy or triple treatment, were administered from the studies mentioned above. Cranial irradiation was excluded from the standard treatment and is now only recommended for children with CNS involvement. 

### 2.3. Development of Minimal Residual Disease (MRD) Diagnostics

In parallel to the intensified therapy, the relevance of genetic risk groups and treatment response became obvious. Improved techniques, such as multicolor immunophenotyping and quantitative PCR, allowed a more precise response detection. 

Detection of MRD by multicolor flow cytometric immunophenotyping started during the 1990s. The leukemic blasts were selected via different antigens (CD45 and CD34, CD117, CD13, CD15, CD33, etc.). 

Langebrake et al. defined different response measurements within AML subtypes, including the prognostics relevance [51]. In contrast, in the SJCRH AML02 study, a higher incidence of relapse was noticed in patients with MRD 1% after the first induction and >0.1% after the second induction. MRD was characterized as a poor prognostic factor EFS and OS [52]. 

In the NOPHO 2004 study, a difference in the EFS, but not in OS, comparing the MRD positive with the morphologic positive patients was noticed [53]. 

Within the Dutch Childhood Oncology Group (DCOG) ANLL, 97, and the MRC 12 trials, MRD levels measured by flow were prognostically favorable for the patients achieving MRD negativity after the first/second induction [54].

Polymerase chain reaction (PCR) is also used to detect MRD for different fusion transcripts. The correlation of expression with the clinical progress of the patients is supported in many studies. The data presented are promising for using this method as a sensitive diagnostic tool. The studies include a small number of patients for the specific subpopulations in the pediatric population, such as t(8;21), in(16) [55].

## 3. Present

The cooperative trial groups achieved significant improvements in overall survival. Table A1 (Appendix A) summarizes recent results, showing similar despite different chemotherapy schemes. 

The analysis of the AML-BFM trials between 1993 and 2010 revealed a continuous improvement of OS but limited progress of EFS [50]. This suggests that 2nd line treatment plays a relevant role in explaining the increasing gap between EFS and OS [56]. 

Treatment regimens include initial double induction and 2- or 3-consolidation blocks. Even if the definitions vary, it is evident that about 60% of children with AML can be cured by chemotherapy only. On the other hand, a significant achievement within the ongoing trials was the establishment of a risk group dependent indication for alloHSCT in the 1st CR [4]. 

Along with the improvements in the transplant procedure and the option to rescue refractory AML, EFS improved [8]. A condition is supportive care to prevent and manage expected complications and more precise diagnostics to allow a genetic and response-based stratification [4]. 

### 3.1. alloHSCT

The AML-BFM Study group and the NOPHO proved that alloHSCT in the 1st CR of high-risk pediatric AML improves EFS, which is not significantly lower than intermediate-risk (IR) or standard risk (SR) [7,8]. However, due to the intensified therapy, including alloHSCT, the salvage treatment is ineffective, compared to IR/SR, resulting in a still inferior OS. 

In case of relapse, all patients indicate alloHSCT in the 2nd CR. Although the prognostic characteristics of relapsed AML are impaired, the survival rate has been maintained or improved [56,57]. 

This allows treating the “right” patient group with the “right” intensity. In particular, the precise definition of the high-risk group by genetics and response associated with the indication of alloHSCT in the 1st CR eliminated significant differences in EFS. 

To achieve this, the results of the AML-BFM 2004 trial have been re-analyzed. Based on the genetic characteristics and augmented by the treatment response to the 1st and 2nd induction, three risk groups could be defined [58]. 

In the AML-BFM 2012 Registry, this risk group stratification has been implemented. The NOPHO Group and others have published similar reports [7]. 

Figure 2 shows the improvement of the HR Group in the AML-BFM 2012 Registry, including the suspension of significant differences between the risk groups.

In addition to the improved risk group stratification, the selection of conditioning regiments, preparation of the transplanted stem cells, the donor identification and availability, and the graft-versus-host prophylaxis significantly contributed to a better outcome in the 1st/2nd CR but also in children with a refractory AML [59]. Within the de-novo AML patients who were transplanted in the 1st CR, the OS increased continuously between 1981 and 2019, documenting the improvement of the treatment approach over time (Figure 3).

Most groups have accepted the standard for myeloablative conditioning with busulfan, melphalan, and cyclophosphamide. Earlier studies with less intensive conditioning (busulfan/cyclophosphamide) resulted in unacceptably high relapses [37]. However, concerns about severe acute toxicities, especially in adolescents, supported the application of alternative regimens, such as treosulfan, fludarabine, and thiotepa [60,61]. Other regiments include clofarabine, busulfan, and fludarabine [59]. 

Regarding stem cell selection, the CD3/CD19-depleted graft transplantation of bone marrow or apheresis cells is the most widely used approach. The donor selection included matched sibling donors (MSD) or matched unrelated donors (MUD), defined as a 9/10 or 10/10 allele match for the HLA loci A, B, C, DR, and DQ, as determined by molecular 4-digit high-resolution typing. For the HR patients without a matched donor, a haploidentical donor is accepted [59]. 

An unexpected, good outcome has been achieved in children with refractory AML, who got fludarabine/amsacrine (FLAMSA) and reduced conditioning (TBI/DLI). The reported 4-years EFS of 41% seems to be promising because, in the past, almost all patients of this cohort died [59].

### 3.2. Diagnostics

The diagnostic of pediatric AML requires morphology, immunophenotyping, and comprehensive cyto- and molecular genetics of the leukemic blasts. All available methods, such as multicolor flow with at least eight colors, panel-next-generation sequencing (NGS), and RNA seq, must be integrated. In general, the risk groups definition can be based mainly on genetics augmented by response measurement by flow and morphology (AIEOP/AML-BFM/FRANCE/UK/COG/Japan), or visa-versa, preferentially MRD-driven augmented by genetics (NOPHO) [62].

Table 1 shows the definition of risk groups according to genetics aberrations and response. The rarity and, in several cases, cryptic translocations require high qualification of the reference laboratories to provide reliable results within a short time frame.

Today, measurement of residual disease by immunophenotyping is the most appropriate method to define initial treatment response. The ongoing treatment protocols use residual disease detection either by immunophenotyping only or in combination with morphology for treatment stratification. The different response kinetics of fusion genes (KMT2A; AML1/Eto, CBL/M) and mutations (NPM1, FLT3-ITD, WT1), measured by quantitative PCR, makes this approach suitable and prognostically relevant only in some subgroups, such as PML/RARA. However, continuous monitoring after remission allows the early detection of molecular relapse. Although it is not entirely proven yet, the treatment of molecular relapse might be feasible with less intensive chemotherapy as bridge to transplant option. The international AmoRe 2017 trial (conducted by GPOH as sponsor) should allow alloHSCT without toxic re-induction in children with a molecular relapse by applying the epigenetically-effective low dose azacytidine. A reduction of MRD -levels to less than 10^−3^ should allow direct alloHSCT.

### 3.3. Myeloid Leukemia of Down Syndrome (ML-DS)

Until almost the end of the 20th century, patients with AML and Down syndrome (DS) were treated identically with the whole group of pediatric AML [63]. In the NOPHO AML-93, after the same treatment, a better 5-year survival was obsereved [64]. In CCG Studies 2861 and 2981, a significantly better 4-year-EFS and no benefit of the BMT was achieved in the patients with DS [65]. A significantly lower relapse rate was noticed in the MRC AML10 study and the BFM-83 and 98 studies [66,67]. Consequently, the therapeutic schema was modified to minimize the toxicities for this favorable group. In the AML02 Trial (JCCSG) patients with ML-DS were treated separately, and a stratification depended on the response after the inductions were implemented [68,69]. In the MRC AML 12, they were allocated for only four courses of chemotherapy and were not eligible for alloHSCT [70]. In the AML-BFM-93 study, the treatment of the DS patients included reduced doses of anthracycline and no high-dose cytarabine/mitoxantrone or cranial irradiation [71]. Since then, these patients have been treated with lower doses of chemotherapy. No maintenance therapy is recommended. Contrary to the excellent response and survival in the case newly diagnosed ML-DS, the overall survival after a relapse, which affects less than 10% of those patients, remains disappointing [65,72].

### 3.4. Supportive Care

All the improvements of the recent decades would be impossible without the progress in supportive care. The introduction of prophylactic antimycotic and, effective antibiotic regimens as well as improved intensive care, including sufficient, sensitive, and specific microbiologic diagnostics, enables intensive treatment with a limited rate of treatment-related death and toxicity [73,74,75]. In addition, recent data confirmed that strict separation of children under immunosuppressive therapy might not be required. The best strategy could be to react immediately with a very high level of awareness and structures in pediatric oncology sites [76]. Unfortunately, these structures are only given in some developed countries. 

### 3.5. Long-Term Toxicities

Pediatric AML and intensive treatment are associated with relevant long-term sequelae. All organ systems could be involved. Although much attention has been spent on cardiotoxicities, especially anthracycline-induced cardiomyopathy, severe damages of the liver, renal function, and endocrinology must be considered. Recent data showed the increased risk of early-onset cardiovascular diseases, neurology, and mental diseases [68]. In addition, treatment-induced malignancies occur in 2 to 5% within 10 to 20 years post-treatment. Unfortunately, to date, there is no plateau of the cumulative incidence [77].

## 4. Future

The therapy of pediatric AML is still based on intensive chemotherapy and, if necessary, alloHSCT. Despite significantly improved survival chances, this therapy has severe acute and long-term side effects [68,78,79]. Therapy-related toxicity is also relatively high at 2 to 4% [75] The aim of new, innovative therapies must be a more targeted treatment, presumably with fewer side effects, without jeopardizing achieved the results. Another aspect is that cure has a the highest priority. While, in adult disease with an age peak above 70 years, it may be beneficial to gain control of the disease for several years, cure must remain the primary aim in children. 

Early deaths from AML in children and adolescents continue to be a significant problem [80,81,82]. While, in some cases, the course is fateful due to the disease dynamics, on the other hand, a higher awareness of pediatricians, general practitioners, and pediatric hospitals could rescue some children. This is especially true for APL and monoblastic leukemia. These must be considered acute emergencies and treated immediately in cooperation with an experienced pediatric oncology center. Effective, quality-assuring structures (central consultation, reference laboratories), as established in some European countries, improve the chances of survival [83].

Several achievements will allow more reliable and precise diagnostics, mainly based on NGS, genome mapping, RNA seq, acetylation/methylation assays, and molecular single-cell characterization [84,85,86]. The challenge will be to integrate the complex data into meaningful results, allowing clinical decisions, better stratification, and more precise treatments. The reliable measurement of MRD by NGS-based approaches will cover all patients and give better insights into the fate of leukemic stem cells, clonal hierarchies, and evolution [86,87,88,89].

Regarding more precise treatment options, the differentiating therapies with all-trans retinoic acid (ATRA) and arsenic trioxide (ATO) in APL are already in use. For the first time, this allows the cure without chemotherapy and with significantly reduced side effects [90]. 

The use of antigen-mediated therapies was successful, especially with the CD33-specific and ozogamicin-coupled antibody GO (Mylotarg) [91]. However, even the positive randomized trials in adults and children have not yet led to a general marketing authorization in pediatric AML [5]. Other specific-acting agents, such as FLT3 or IDH1/2 inhibitors, have also shown efficacy in children and adolescents. Still, it has not been conclusively investigated whether this improves the chances of a cure or only means an effective but transient blast reduction [61]. The same is probably true for epigenetic approaches, which have been used very successfully in older adults. Combining BCL-2 inhibitors (venetoclax) with low-dose, mainly epigenetically active chemotherapy (e.g.azacytidine) modifies the clinical course of myelodysplastic syndromes/AML, especially with low proliferation activity, very positively with significantly improved survival [92]. Other approaches combined venetoclax with MDM2 or FLT3-ITD inhibitors; however, experience in children is lacking. The addition of venetoclax to high-dose cytarabine with or without idarubicin revealed a promising overall response of 69% [93]. Nevertheless, the contribution of venetoclax to this response rate needs to be evaluated. In summary, the relevance of venetoclax in pediatric AML needs to be confirmed in further trials (such as planned within the LLS-PedAL initiative). 

Another complex area of AML therapy includes various immunotherapies. One approach will be post-HSCT immunomodulation. The effectiveness has already been shown in post-HSCT treatment with donor-lymphocyte infusions (DLI) but definitively provides more options. Cytokine-induced killer (CIK) cells have already been introduced to clinical trials [94]. Other approaches use activated NK-cells [95]. In particular, the combination with immunomodulatory agents that optimize cellular treatments shows promising efficacy in preclinical and initial clinical studies. Directly related to these approaches are the current research findings on the importance of the microenvironment [96], its interaction with leukemic blasts, and the effects it induces on the selective proliferation of malignant cells, support of escape mechanism of leukemic stem cells, and inhibition of immunocompetence of effector cells, such as T/NK cells [97].

The successful cellular therapy approaches in B-cell lymphocytic leukemia/lymphomas raise high hopes for myeloid neoplasms [98]. However, the challenge in pediatric AML is more complex, while it is relatively easy to compensate for B lymphocyte eradication with antibody substitution, the reconstitution of myelopoiesis is only feasible with alloHSCT. Accordingly, to date, these therapeutic options must be viewed primarily as “bridge-to-transplant” regime. Nevertheless, cellular treatment options, such as gene-engineered T-CAR or NK-CAR cells, are promising approaches to enable more precise and hopefully less side-effective therapy in the future [99,100,101] Several targets have been addressed so far (CD33, CD123, CLL1, and others) [101,102,103]. 

Overall, it is unlikely that there will be “the one” effective therapy for a heterogeneous disease like AML. Only the optimized combination of all available options will allow a further, significant improvement of cure rates so that likely different treatments adapted to the AML subtype are needed. This underlines the need for further comprehensive research into the mechanisms of leukemogenesis, specific therapies, and, above all, systematic clinical research to develop scientifically-validated treatments, despite the small number of cases. This will only be possible if the international collaboration between the study groups will be further improved on a global level. It is important to learn more about small subgroups and more precise treatment, as well as reducing inequalities between countries and continents. The recently established Leukemia & Lymphoma Society (LLS), Pediatric Acute Leukemia (PedAL) initiative and the European Pediatric Acute Leukemia (EuPAL) foundation are on the way to launching such a platform in North America, Europe, Australia, and, hopefully, in Japan [3,104].

In this context, the therapies of AML in children, even if they have low economic impact, should not be considered exclusively as a “waste product” of adult medicine but should have a right to their own, child-specific therapy development. This applies to both, research on pediatric therapies and the timely establishment of therapies that have been successfully used in adult AML.

## Figures and Tables

**Figure 1 jcm-11-00504-f001:**
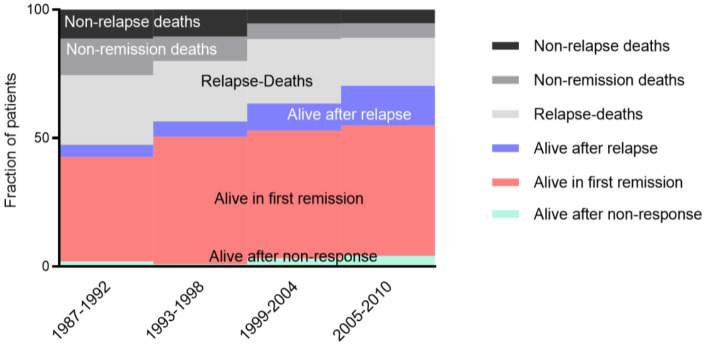
Improvement of outcome in pediatric AML. Continuous increase of survival in first remission (red), following initial non-response (green), and after relapse (blue). In parallel, the treatment-related mortality (black, non-relapse deaths) and non-remission deaths (grey) decreased significantly. This supported the hypothesis that the improved overall survival is based on better treatment and improved supportive care.

**Figure 2 jcm-11-00504-f002:**
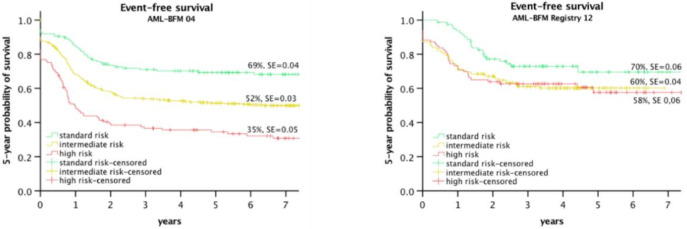
EFS of risk groups in Study AML-BFM 2004 [58] and AML-BFM registry 2012 [8].

**Figure 3 jcm-11-00504-f003:**
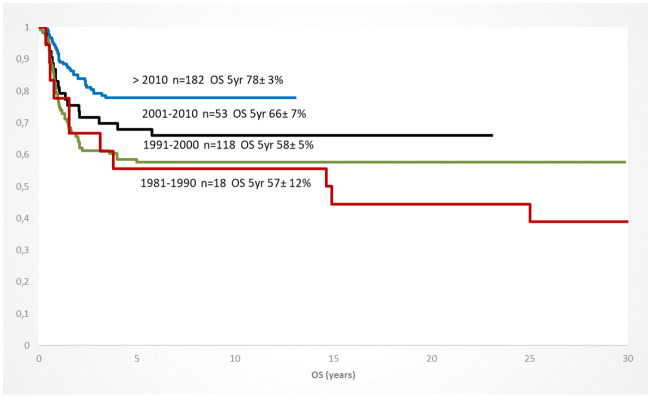
Increase of overall survival of children with AML and HSCT in CR1 (any donor/conditioning) since 1981; 10 year-periods (AML-BFM trials).

**Table 1 jcm-11-00504-t001:** Risk Group definition by genetics and response, an example from the AIEOP-BFM AML 2020 Study.

Risk Group	Genetic Risk Criteria	Response Criteria
Standard Risk (SR)	CBFβ abnormalities t(8;21)(q22;q22) with adequate (≥2 log) reduction by qPCR at IND2 inv(16)(p13q22)/t(16;16)(p13;q22) Biallelic CEBPα aberrations t(16;21) CBFA2T3/RUNX1 *and* FLT3-ITD negative	Genetic standard risk and MRD < 0.1% at IND 2 t(8;21) and MRD > 2 log reduction at IND 2 (qPCR)
Intermediate Risk (IR)	NON SR and NON HR patients	Genetic standard or intermediate risk and MRD at IND 1 ≥ 0.1% and <1% and MRD at IND 2 < 0.1%
High Risk (HR)	Complex karyotype (≥3 aberrations including at least one structural aberration) *excluding those with recurrent translocations* Monosomal Karyotype, i.e., -7, -5/del(5q) 11q23/KMT2A rearrangements involving: t(4;11)(q21;q23) KMT2A/AFF1 t(6;11)(q27;q23) KMT2A/AFDN t(10;11)(p12;q23) KMT2A/MLLT10 t(9;11)(p21;q23) KMT2A/MLLT3 with other cytogenetic aberrations t(16;21)(p11;q22) FUS/ERG t(9;22)(q34;q11.2) BCR/ABL1 t(6;9)(p22;q34) DEK/NUP214 t(7;12)(q36;p13) MNX1/ETV6 inv3(q21q26)/t(3;3)(q21;q26) RPN1/MECOM 12p abnormalities$break$ FLT3-ITD with AR ≥ 0.5 not in combination with other recurrent abnormalities or NPM1 mutations WT1 mutation and FLT3-ITD inv(16)(p13q24) CBFA2T3/GLIS2 t(5;11)(q35;p15.5) NUP98/NSD1 and t(11;12)(p15;p13) NUP98/KDM5A Pure Erythroid leukemia	MRD ≥ 1% at IND 1 or ≥0.1% at IND 2 or (only if FLOW-result not available/informative) blast count ≥5% at IND 1

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
