# Peer review of "Pediatric Acute Myeloid Leukemia—Past, Present, and Future"

_jcm, 2022, doi:10.3390/jcm11030504_

Round 1
Reviewer 1 Report
This is comprehensive review concerning pediatric acute myeloid leukemia.
The authors described in details the history of the development of pediatric AML treatment including progress in diagnostics, risk groups stratification, chemotherapy, hematopoietic stem cell transplantation and supportive care.
Data on the treatment results from the main study groups from Europe, North America and Japan were presented as well as current treatment options.
Finally future perspectives were described like precise treatment and need for further international collaboration between the study groups on a global level.
Specific comments
Table 1
The most recent references concerning the MRC (Burnett A.K Br J Haematol, 2020, 188: 86-100), PPLLSG (Czogala M, Cancers. 2021; 13(18):4536) and SJCRH (Rubnitz JE J Clin Oncol.2019;37(23):2072-2081) were not included in the table. I would suggest to include the most current data especially if the newly reported results differ from those included in the table.
Line 231, 245
Error! Reference source not found. – Please clarify.
Author Response
We would like to thank the reviewer for the positive comments. The suggested improvements have been integrated.
Reviewer 2 Report
This review paper is very good and useful.
A bit more about innovative treatments or salvage therapies could be added.
Maybe the author could speculated which approaches used in adults may be ar some stage more widely used in children.
I would like the authors to discuss venetoclax more extensively. There are some promising new data such as:
Venetoclax in combination with cytarabine with or without idarubicin in children with relapsed or refractory acute myeloid leukaemia: a phase 1, dose-escalation study Seth E Karol, MD. Lancet Oncology 2020.So venetoclax with chemotherapy and not only azacitidine should be discussed.
Author Response
We would like to thank the reviewer.
We added some innovative treatment approaches in section "future".
We also discussed the combination of venetoclax with intensive chemotherapy
Reviewer 3 Report
In their review report titled "Pediatric Acute Myeloid Leukemia- Past, Presence, Future" Reinhardt D. and colleagues attempted to historically review the milestones in the field of pediatric acute myeloid leukemia. They approached some issues such as diagnostic, treatment, risk groups, and outcome. Ultimately they aimed to give a short future perspective. Basically, the current manuscript is poorly readable for a couple of reasons that will be hereafter shortly summarized: 1) the language requires massive amelioration; 2) being a review, one expects a greater spirit of criticism that is currently missing. Too often the manuscript looks like a mere shallow historical report and appears still in a drafting stage, rather than a finalized paper. Few examples are given: tables should be indicated within the main text with their correct numbering, instead, they are not (with the only exception of table 1 for which the authors indicate "The following table summarize ......" line 192). The same for figures. Even if there is only a single one. Concerning this latter point, I do not understand why there are 3 risk groups but in the legend, there are 6! In a Kaplan-Meier plot, all members' groups must be censored, otherwise is no way to draw such curves! Is it not? Thus, there is no need to indicate uncensored members. Or am I missing something? In addition, a couple of times throughout the manuscript it appears "Error! Reference source not found". What do they mean? Ultimately, references should be carefully checked and formatted accordingly to the journal policy. The same for acronyms, when they are indicated for the first time they should be followed, in extenso, by their meaning.Author Response
Improvement of the language has been made.
Concerning the figure, there are and remain 3 risk groups. The other information in the legend only explains the censored patients, as requested by the reviewer.
All information and references have been checked
Reviewer 4 Report
The authors report pediatric acute myeloid leukemia.
- The authors should provide the flow chart data (i.e. timeline) in a Figure.
- The authors should summarize the clinical trial list in a Table.
- The authors should provide the representative chemotherapy regimens in a Figure. It will be benefit for the reader.
Author Response
We thank the reviewer, the language has been improved.
The different clinical trials are presented in a table. Due to the varieties of the treatment protocols, a single figure will not allow giving an overview.
Reviewer 5 Report
In the manuscript, “Pediatric acute myeloid leukemia- past, present, and future,” the authors have described the development of pediatric AML with a representation of past, present, and future diagnosis, treatment, and outcome of the patients. The authors have clearly outlined the review by explaining the specifics of the clinical trial in the past and present times and a feasible treatment option in the future for pediatric AML.
However, in the introduction, the authors should add some background about the pediatric AML characteristics, focusing on cytogenetics, cause, and prognosis.
In the introduction, with the help of databases like TCGA, authors could explain the prognosis and survival of AML by generating a graph.
In sections 4.2 and 4.3, the authors need to add the citations for the data and statements mentioned.
In section 5, the authors mentioned engineered T cell therapy as the activated NK cell, which is incorrect.
Authors need to check for typographical and grammatical errors. For: e.g., In section 2.1, 80th has to be replaced by the 80s.
Author Response
We thank the reviewer for the great suggestions. Some background information has been added as well as supporting citations. We also integrated an exemplary figure showing the improvement of prognosis overtime.
Misspellings etc . have been corrected.
Round 2
Reviewer 3 Report
Though I partly appreciated the improvements made, the revised version of the manuscript has not been strengthened sufficiently. Remarkably, besides some spotted criticisms, the authors did not answer accurately to the reviewer's concerns. It is understandable that they might disagree with a reviewer. However, they should provide a clear response and not very few short and scant sentences. Noticeably, they claim that the references have been checked, but in the current version, they are still not properly formatted in accordance with the journal guidelines. Eventually, concerning new Figure 1 and Figure 3, I do not understand how they have been made, and by checking the references (#3, #4, and #59) it looks that they are not reproduced from those articles. Are they the outcome of a meta-analysis? If so it should have been detailed. Concerning Figure 2, I consider it a little bit unusual because it displays Kaplan-Meier/survival curves of 3 different cohorts that must have been censored, but in the plot inlet, they also indicated the uncensored ones. My question is simple: what is the meaning, and the difference, among the cohorts (e.g. standard risk vs standard risk-censored)? how is it possible to draw a survival curve if the participants are uncensored? That's a little bit confusing for the readers. Furthermore, it would have been appreciable if they could have provided a revised version with the changes highlighted in some way, either in the main-text file or, alternatively, in a more detailed point-by-point rebuttal letter.
Author Response
We thank the reviewer for the comment, which we will try to answer:
Reply to the comments:
"Though I partly appreciated the improvements made, the revised version of the manuscript has not been strengthened sufficiently. Remarkably, besides some spotted criticisms, the authors did not answer accurately to the reviewer's concerns. It is understandable that they might disagree with a reviewer. However, they should provide a clear response and not very few short and scant sentences."
We have tried to consider the various criticisms and suggestions of the different (5) reviewers, although we were not successful in all cases, especially when recommendations contradict each other.
1st review: The main criticisms were 1) the language requires massive amelioration (2) being a review, one expects a greater spirit of criticism that is currently missing.
Do we miss something? What are the reviewer's expectations of a review? To judge retrospectively about decisions in the past?
We wanted to summarize the most essential steps and developments of the last decades as we understand it. The last part can give some ideas for promising directions and necessary structures.
While some of the hypotheses tested in the earlier studies failed, overall survival improved significantly in the approaches conducted by cooperative study groups, whether conducted in North America, Europe, or Japan. International collaboration has contributed to this success. (see Zwaan et. JCO).
Too often the manuscript looks like a mere shallow historical report and appears still in a drafting stage, rather than a finalized paper.
This is an opinion.
Noticeably, they claim that the references have been checked, but in the current version, they are still not properly formatted in accordance with the journal guidelines.
We used the provided templates, if it does not fit with the guidelines, we will modify it according to the editor's request.
Eventually, concerning new Figure 1 and Figure 3, I do not understand how they have been made,
The figure 1 is based on the AML-BFM data reported in the manuscript Rasche et al. Leukemia 2018. The better description or origin has been added.
Figure 3 is an actual analysis of the AML-BFM database, following and documenting all children and adolescents with the diagnosis of AML since 1978. Again, the appropriate description has been added/ maybe better explained.
and by checking the references (#3, #4, and #59) it looks that they are not reproduced from those articles. Are they the outcome of a meta-analysis? If so it should have been detailed.
Text in the document: "Except for acute promyeloblastic leukemia (APL), improved survival has been achieved by using long-known conventional drugs, mainly cytarabine and anthracyclines. Scheduling risk group stratification, modifications of allogeneic stem cell transplant (alloHSCT), and management of complications allowed the cure of most children. " 3/4
Reference #3 describes the establishment of the international PDCD. The basis/objective is, on the one hand, the lack of progress in therapy and, among other things, the identification of new risk groups/therapy options by using existing data resources.
Verbatim quote: "Despite the rise in cure rates for children with cancer over the past 60 years, many children die from their disease, especially in middle- and low-income countries, 12 and many suffer long-term side effects." Plana et al #3
The work of Zwaan et al. (#4) describes international efforts to find and harmonize common treatment strategies. These concerns risk groups, subgroups (e.g. APL), treatment regimens, supportive care, and others.
"There is general agreement across groups about the definition of high-risk (HR) AML. Groups differ in the use of low, standard, and intermediate risks to designate all other types of AML. For the purposes of this review, disease in all patients with non-HR AML will be called standard-risk (SR) AML." Zwaan et al. #4
"Without doubt, the advancements in supportive care have contributed significantly to the improvement in outcome for pediatric patients with AML, given the requirement for intensive chemotherapy that is needed to treat AML." Zwaan et al #4
Reference #59 is cited 4x.
Manuscript "In addition to the improved risk group stratification, the selection of conditioning regiments, preparation of the transplanted stem cells, the donor identification and availability, and the graft-versus host prophylaxis significantly contributed to a better outcome in 1st /2nd CR but also in children with a refractory AML" 59
Statements in the Sauer manuscript:
"Therefore, the trial was amended to allow for matched donor HCT in CR-1 for those children presenting defined molecular and cytogenetic high-risk constellations" Sauer et al #59
"Since emerging data in the early 2000s hinted at the importance of offering early allo-HCT to this group, AML responding poorly to treatment was an indication for early HCT. Originally, reduced intensity regimens were known to be better tolerated, however, dose intensity had been shown to influence leukemic control directly" Sauer et al #59 .
In AML SCT-BFM 2007, 4-year "EFS for newly diagnosed poorly responding disease reached 53% as analyzed by an "intent-to-treat" approach, whereas poorly responding relapses were more difficult to rescue, with a 4-year EFS reaching 29% only."
Text in the manuscript: "Regarding stem cell selection, the CD3/CD19-depleted graft transplantation of bone marrow or apheresis cells is the most widely used approach. The donor selection included matched sibling donors (MSD) or matched unrelated donors (MUD), defined as a 9/10 or 10/10 allele match for the HLA loci A, B, C, DR, and DQ, as determined by molecular 4-digit high-resolution typing. For the HR patients without a matched donor, a haploidentical donor is accepted"
"For the majority of patients, an HLA-matched donor, either related or unrelated, was a prerequisite for enrollment. In case a matched donor could not be identified, a haploidentical HCT was accepted for children with poorly responding disease only. HLA matching was defined as a 9/10 or 10/10 allele match for the HLA loci A, B, C, DR, and DQ, as determined by molecular 4-digit high-resolution typing. In the absence of an HLA-matched donor, transplantation from a haploidentical donor was accepted if the child was diagnosed with poorly responding AML. "Sauer et al. #59
In this respect, the citations support the statements.
Nevertheless, we added the Creutzig Blood publication1 which addresses all the mentioned topics.
Concerning Figure 2, I consider it a little bit unusual because it displays Kaplan-Meier/survival curves of 3 different cohorts that must have been censored, but in the plot inlet, they also indicated the uncensored ones. My question is simple: what is the meaning, and the difference, among the cohorts (e.g. standard risk vs standard risk-censored)? how is it possible to draw a survival curve if the participants are uncensored? That's a little bit confusing for the readers.
In Figure 2, 3 Kaplan-Meier curves are shown for each of the SR, IR, and HR groups. In the legend, the respective line/level is described with a color (green = standard risk). The legend "standard curve censored" only describes that a short vertical line indicates the event that was censored.
Furthermore, it would have been appreciable if they could have provided a revised version with the changes highlighted in some way, either in the main-text file or, alternatively, in a more detailed point-by-point rebuttal letter.
We hope that we addressed at least some of the points raised by the reviewer.
References
- Creutzig U, van den Heuvel-Eibrink MM, Gibson B, et al: Diagnosis and management of acute myeloid leukemia in children and adolescents: recommendations from an international expert panel. Blood 120:3187–205, 2012 (16)

Reviewer 4 Report
N/A
Author Response
Thank you